# Proteomic Analysis of Apple Response to *Penicillium expansum* Infection Based on Label-Free and Parallel Reaction Monitoring Techniques

**DOI:** 10.3390/jof8121273

**Published:** 2022-12-03

**Authors:** Meng Xu, Kaili Wang, Jun Li, Zhuqing Tan, Esa Abiso Godana, Hongyin Zhang

**Affiliations:** 1School of Food and Biological Engineering, Jiangsu University, Zhenjiang 212013, China; 2Analysis & Testing Center, Jiangsu University, Zhenjiang 212013, China

**Keywords:** apple, label-free, *Penicillium expansum*, defense response

## Abstract

Blue mold, caused by *Penicillium expansum*, is the most destructive fungal disease of apples and causes great losses during the post-harvest storage of the fruit. Although some apple cultivars are resistant to *P. expansum,* there has been little information on the molecular mechanism of resistance. In this study, differential proteomic analysis was performed on apple samples infected and uninfected with *P. expansum*. Parallel reaction monitoring (PRM) technology was used to target and verify the expression of candidate proteins. The label-free technique identified 343 differentially expressed proteins, which were mainly associated with defense responses, metal ion binding, stress responses, and oxidative phosphorylation. The differential expression of enzymes related to reactive oxygen species (ROS) synthesis and scavenging, the activation of defense-related metabolic pathways, and the further production of pathogenesis-related proteins (PR proteins) during *P. expansum* infection in apples, and direct resistance to pathogen invasion were determined. This study reveals the mechanisms of apple response at the proteomic level with 9 h of *P. expansum* infection.

## 1. Introduction

Apple is a widely consumed fruit, rich in vitamins, dietary fiber, and many other nutrients [1]. During post-harvest storage, apples are susceptible to a variety of pathogens, including blue mold, grey mold, and black spot disease. Blue mold caused by *P. expansum* is the most widespread and damaging disease, causing serious economic losses to the apple industry. Patulin (PAT) produced by *P. expansum* also poses a great risk to human health [2]. Chemical fungicides are often used for disease control, but due to environmental concerns, safer and more convenient control methods are increasingly sought. Therefore, an in-depth study of the mechanisms of action between *P. expansum* and apple could help find more suitable methods for disease control.

When plants are exposed to pathogens, a complex response is generated to mount an immune response. The plant cell wall is the first barrier against pathogens, and when pathogens break through it, they are recognized by pattern receptors on the plant surface, which activate pathogen-associated molecular pattern (PAMP)-triggered immunity (PTI) and trigger many downstream responses. Plant pattern recognition receptors (PRRs) are located on the cell surface, including plasma membrane-bound receptor-like kinases (RLKs) and receptor-like proteins (RLPs). PTI has a broad-spectrum defense role and is not sufficient to fully resist pathogen invasion. The pathogen releases effectors to overcome PTI, and the plant resistance (R) protein senses pathogen-associated avirulence basic (Avr) proteins to trigger the expression of the physiological defense response in the plant, called effector-triggered immunity (ETI). To defend against pathogenic bacteria, both PTI and ETI induce the production of a range of antimicrobial peptides, pathogenesis-related (PR) proteins, and other physiological defense substances [3].

Some good developments has been made in extending the research on the interaction of the molecular mechanisms between *P. expanusm* and apple. The transcription factor CreA (a global regulator of carbon catabolism) was assessed to be toxic and associated with PAT synthesis both in vitro and in vivo; knocking out *CreA* resulted in *P. expansum* being virtually nontoxic and not producing PAT, and these mutants could not be successfully colonized on apple [4]. *PacC* plays an important role in fungal adaptation to environmental pH. Chen et al. [5] found that *PePacC* can act as an effector for a variety of target proteins and plays an important role in extending the virulent synthesis of *P. expansum* [5]. Prusky et al. [6] found that *PePG1* expression is strongly correlated with environmental pH on apples and citrus, and the environmental acidification plays an important role in increasing the pathogenicity of pathogenic fungi [6]. Levin et al. [7] revealed 18 possible genes encoding the LysM protein for the first time and studied the regulatory role of LysM in the spread of *P. expansum* infection by gene knockout and other techniques [7].

Proteomic studies have shown their importance in studying plant–pathogen interactions. Unlike genomic tools, proteomics helps to understand the identity, abundance, turnover, post-translational modifications (PTMs), and interactions of different proteins in a mixture. Yang et al. [8] compared wheat inoculated and uninoculated with *Puccinia striiformis* f. sp. *tritici* (*Pst*) by quantitative proteomics and showed that 530 proteins were differentially expressed and several proteins were involved in the response to *Pst* infection [8]; Ravi et al. [9] compared the proteomes of *Xanthomonas oryzae* pv. *oryzae* (*Xoo)*-sensitive and -resistant rice plants and analyzed the differential proteins and screened 23 defense candidate proteins to improve resistance to *Xoo* by overexpressing arginase 1(*OsArg1*) in sensitive plants [9]. Lu et al. [10] compared leaf blight (LB)-resistant and LB-sensitive plants of orchids by proteomics and identified proteins in the reactive oxygen species metabolic pathway and found that Cu/Zn superoxide dismutase (BsSOD1) protein abundance and its gene expression were higher in LB-resistant plants than in LB-sensitive plants [10]. Proteomics can tell us what is happening in a particular state of a cell, a tissue, or an organism and what the protein–protein interactions are. Changes that occur within specific tissues and cells can be better predicted, revealing their mechanisms of action [11].

In this study, we used high-throughput proteomics to analyze the changes in apple protein levels before and after *P. expansum* infection and bioinformatics tools to analyze the differential proteins and investigate the defense mechanisms of apples during *P. expansum* infection. This study provides some theoretical references to the molecular mechanisms of apple defense against *P. expansum* infection.

## 2. Materials and Methods

### 2.1. Pathogen

The *P. expansum* strain was isolated by our research team (maintained at the China Center for Type Culture Collection and is numbered as CCTCC AF 2022039). The activated strain was incubated on PDA medium at 25 °C for seven days before the experiment. Then the spores were collected and adjusted to a concentration of 1 × 10^8^ spores/mL in 0.9% saline.

### 2.2. Fruit

Apple (*Malus domestica*) fruits cv red Fuji at commercial maturity with uniform shape and size and no pests, scars, or mechanical damage (fruit hardness 7 N/cm^2^, 13% soluble solid material) were selected as the test object. The selected fruits were soaked in 0.2% sodium hypochlorite solution for 2 min and then carefully rinsed with tap water for 5 min to wash off the surface dirt, as well as the sodium hypochlorite solution, and then air-dried.

### 2.3. Determination of Sampling Time Points and Preparation of Apple Samples

After the apples were air dried, three wounds of size 5 mm × 4 mm were punched evenly along the equatorial line of the apples with a hole puncher. After drying for 20 min, the prepared *P. expansum* spore suspension was injected into the wounds with a pipette, 30 μL per wound, and the control group was injected with the same volume of saline. The apples were placed evenly in clean frames, one frame for every 12 apples, wrapped in cling film, and placed in an incubator at 25 °C and RH 95%. Then, at 1 h, 3 h, 6 h, 9 h, 12 h, and 24 h after inoculation with molds, samples were taken at the wound site with a sterile scalpel, and the germination of mold spores was observed under a microscope after staining and preparation with lactate phenol cotton blue staining solution, so as to determine the best sampling time point. Then sampling was carried out at the best sampling time point. After the decaying part of the wound was cut with a sterile scalpel, 2–3 mm flesh was taken as a sample and quickly placed into liquid nitrogen. The samples were then stored in a refrigerator at −80 °C for further experiments.

### 2.4. Proteome Sample Preparation

The extraction of apple proteins was according to Tan’s experimental method. Briefly, 5 g of apple tissue was ground in liquid nitrogen, washed three times with 15 mL of cold acetone, dried, and added to 15 mL of extraction buffer (0.1 M Tris pH 8.8, 10 mM ethylenediaminetetraacetic acid (EDTA), 0.4% β-mercaptoethanol, 0.9 M sucrose) and 15 mL tris-saturated phenol (pH 8.8), mixed well on ice and centrifuged at 12,000× *g* for 20 min. Then, the phenol phase was collected in a new tube, repeated once, and the two phenol phases were combined and centrifuged at 10,000× *g* for 15 min to remove the supernatant. The supernatant was incubated with 25 mL of a methanol solution containing 0.1 M at −20 °C overnight, then centrifuged and the supernatant was discarded. The precipitate was washed with acetone three times and air-dried, stored at −80 °C in the refrigerator [12]. The protein precipitate was dissolved with 8 M urea, and the protein supernatant was transferred to a new EP tube; the concentration of the protein was detected by the BCA protein quantification kit, and the integrity of the protein bands was measured by SDS-PAGE. Protein samples were desalted with C18 small columns, and the treated samples were lyophilized with a freeze dryer, stored in a −80 °C refrigerator, and redissolved with 0.1% formic acid water before mass spectrometry analysis.

### 2.5. Proteomic Mass Spectrometry

On-board operation was performed using a Thermo Q-Exactive mass spectrometer with the following parameters: column type: 75 μm × 15 cm, 2 μm particle size, C18, 100 Å, Acclaim PepMap. with trap enrichment column. Mobile phase: A: aqueous solution (0.1% formic acid) B: 100% ACN (0.1% formic acid); detection mode: full MS/dd-MS2; ion source voltage: 2.0 KV; scan range: 350–1800 *m*/*z*; ion source temperature 320 °C; s-lens: 50 °C resolution: 70,000; secondary mass spectrometry: starting fixed *m*/*z* 110. resolution 70,000; secondary mass spectrometry: starting fixed *m*/*z* 110, resolution 70,000; secondary rupture mode: HCD; AGC setting: primary 3 × 10^5^, secondary 1 × 10^5^; dynamic exclusion time: 60 ms.

### 2.6. PRM Verification of DEPs

According to the proteomic results, four proteins were randomly selected to conduct targeted qualitative analysis of the target proteins by LC–PRM/MS. After the peptides were separated by high-performance liquid chromatography, a Q-Exactive HF mass spectrometer (Thermo Fisher Scientific, 81 Wyman Street, Waltham, MA, USA) was used for PRM mass spectrometry analysis. Finally, Skyline 3.5.0 was used to analyze the data of the PRM original files.

### 2.7. Data Analysis

Data from this trial were searched for databases using Maxquant 1.6.17.0 and quantified for LFQ, including the quantification of proteins, peptides, and PSM. The apple protein database was compared with UP000290289 (*Malus domestica*) from UniProt (http://www.uniprot.org/, accessed on 26 November 2021). GO functional annotation was searched on UniProtprot (http://www.uniprot.org, accessed on 24 February 2022); KEGG function was annotated with KOBAS (http://kobas.cbi.pku.edu.cn/, accessed on 5 March 2022). The prediction of the subcellular localization of key proteins was performed using WoLF PSORT (https://wolfpsort.hgc.jp/, accessed on 27 November 2022).

## 3. Results

### 3.1. Result Analysis

#### 3.1.1. Selection of Key Time Points for Sampling

Samples were taken, and wounds were made at different time points during *P. expansum* infection. It was found by microscope observation that 1 h after inoculation with the bacterial suspension, mold spores began to change from an oval shape to a shape with a tip. Then, 3 h after inoculation with the bacterial suspension, spore deformation was obvious, and budding tubes emerged. Then, 6 h after inoculation with the bacterial suspension, the budding tubes of conidia directly invaded the apple tissue, and the tails of some conidia expanded and formed appressorium. After 9 h inoculation with the bacterial suspension, the infecting nails grew on the appressorium and infected the apple tissue. The bud tube extended into the apple tissue and continued to infect the apple tissue. Then, 12 h after inoculation, primary hyphae continued to extend into apple tissues. After inoculation with the bacterial suspension for 24 h, new spores were formed on the secondary mycelia. At 9 h after inoculation, *P. expansum* successfully infected apples, so the sampling time was determined to be 9 h (Figure 1 and Figure 2).

#### 3.1.2. Screening for Differentially Expressed Proteins

A total of 4164 proteins were identified in the *P. expansum* infected and non-infected apple samples. In the quantitative results, the screening for significantly expressed differential proteins was based on FC > 1.5 or FC < 0.67 and *p* < 0.05. The results showed that a total of 343 differential proteins were screened. Among them, 131 proteins were downregulated and 212 proteins were upregulated. Functional annotation and enrichment analysis were performed for all differentially expressed proteins. Figure 3 shows volcano plot of the differentially expressed proteins, where the upregulated proteins are in red and the downregulated proteins are in blue.

#### 3.1.3. Subcellular Localization of Differentially Expressed Proteins

Studying the mechanisms and patterns of protein localization in cells and predicting the subcellular localization of proteins are important for understanding protein structure, properties, functions, and protein interactions. In this study, it can be seen from the prediction results of subcellular localization that the differentially expressed proteins are mainly located in nucleus, cycloplasm, and chloroplast (Figure 4). The upregulated proteins and downregulated proteins have similar subcellular localization. These proteins may interact in these organelles and perform important biological functions.

#### 3.1.4. GO Enrichment Analysis of Differentially Expressed Proteins

The DEPs were annotated into the GO database, and 319 of the 343 differentially expressed proteins were annotated into the GO database for cellular component, molecular function, and biological process classification. Among the cellular components, cytosol, cytoplasm, an integral component of the membrane, and ribosome were the most enriched, indicating that most biological processes in apple defense occur in the cytoplasm and membrane. The subclasses enriched in molecular functions are ATP binding, metal ion binding, oxidoreductase activity, GTP binding, and hydrolysis activity. This suggests that ATP synthesis, metal ions, and some oxidoreductases involved in oxidation and reduction reactions play important roles in the defense response of apple. The subclasses that were more enriched in biological processes were translation, protein transport, tricarboxylic acid cycle, defense response, and protein folding. The GO enrichment of the differential proteins is shown in Figure 5, where only the top 15 pathways were enriched for differential proteins at three levels of classification.

#### 3.1.5. KEGG Enrichment Analysis of Differentially Expressed Proteins

The results of the annotation of the DEPs in the KEGG database showed that the differentially expressed proteins could be annotated into a total of 19 secondary subclasses of six primary classifications (Figure 6), with 11 secondary subclasses common to both the up- and downregulated proteins, of which glycan biosynthesis and metabolism, signal transduction, nucleotide metabolism, replication and repair, and signal transduction were unique to the downregulated proteins. Glycan biosynthesis and metabolism, the metabolism of terpenoids and polyketides, transcription, and signal transduction were unique to the downregulated proteins. Polyketides, transcription, and translation were unique to the upregulated proteins. These specific metabolic pathways may play important functions in the response of apples to extended *P. expansum* infection. KEGG pathways with *p* < 0.05 were considered significantly enriched pathways, and a total of 32 pathways belonging to 11 secondary subclasses were screened. The most enriched KEGG pathway was metabolic pathways, with 61 proteins annotated to this pathway.

When apples sense pathogenic invasion, multiple metabolic pathways are activated in vivo, such as the ubiquitin–proteasome pathway mediating protein degradation; the production of secondary metabolites, such as lignans and flavonoids, to enhance resistance through the phenylpropane biosynthesis pathway and the flavonoid biosynthesis pathway; the activation of the mitogen-activated protein kinase (MAPK) signaling pathway in response to defense plant signals; and the activation of reactive oxygen species production and the promotion of downstream immune protein expression through the plant–pathogen-interactions pathway. Table 1 lists some of the metabolic pathways associated with the apple defense response to *P. expansum*.

#### 3.1.6. Validation of Selected Candidates by PRM

According to the results of the proteomics of apple infected by *P. expansum*, four proteins were randomly selected and analyzed by PRM. The results showed that the expression of the four identified proteins (A0A498JTJ2, A0A498JUX9, A0A498KA74, and A0A498KKS6) was consistent with the trend of proteomic data, indicating that the proteomic results were reliable (Figure 7).

#### 3.1.7. Expression of Pathogenesis-Related Proteins

Pathogenesis-related proteins (PRs) are low-molecular-weight proteins that are selectively soluble at low pH and are mainly found in the vesicles and extracellular spaces. PR proteins are classified into 17 families, PR-1 to PR-17, based on their molecular weight, biochemical function, and properties [13]. Among the differential proteins of apple, a total of nine proteins have been identified as PR proteins (Figure 8), of which four proteins belonging to the Bet-v-1 family and three proteins belonging to the Mal d family with ribonuclease activity belong to PR-10, one β-1,3-endoglucanase belongs to PR-2, and one peroxidase (POD) belongs to PR-9. PR-10 is a highly conserved protein that is induced by biotic and abiotic stresses. PR-10 proteins all have a common fold that acts as a binding site for different small molecules, mainly cytokinins and sterols. Furthermore, it has been shown that different protein ligands affect the stability of Bet-v-1. This dual characteristic provides a plausible explanation for the differential expression of bet-v-1 in apples. PR9 has POD activity and participates in the synthesis of lignin, enhancing the structure of the cell wall and the scavenging of reactive oxygen species in the fruit.

## 4. Discussion

Plants usually mount an immune response in the face of pathogen infection through two defense systems. One is the plant’s immune system, and the other is a pathogen-induced defense system. The cross-response between the two systems triggers the activation of multiple signaling pathways in the plant, changes in hormone levels, and the production of disease-resistant proteins to respond to the infection.

In the present study, the differential expression of many proteins was induced under *P. expansum* infection, and most of these DEPs were associated with plant defense mechanisms against pathogenic invasion. Proteins associated with biological processes, such as the MAPK cascade, plant–pathogen interactions, the stimulation of metabolite biosynthesis, and amino acid metabolism were significantly upregulated in expression. They are involved in various processes such as reactive oxygen species (ROS) burst and scavenging, hypersensitive response (HR), redox reactions, and defense-related protein induction.

The KEGG metabolic pathway shows that DEPs are involved in several metabolic pathways related to plant defense, including MAPK (plant) and plant–pathogen interactions (Figure 9). The pathway diagram shows that pathogen stimulation causes an increase in the expression of calmodulin and enhanced disease susceptibility 1 (EDS1) and heat shock protein 90 (HSP90), which in turn trigger a hypersensitive response in apple, and the accumulation of EDS1, an important signaling molecule in the regulation of plant immunity, is essential for the subsequent amplification of disease resistance signals during pathogen infection [14]. HSP90 is widely distributed in plant cells and is a molecular chaperone protein); in addition, MPK4 expression, was downregulated at 9 h of *P. expansum* infection. It has been reported that MPK4 plays a negative regulatory role in plant immunity and can be activated by PAMP [15,16]; Tereza et al. [17] showed that the Gnomon molecular chaperone protein plays an important role in plant stress resistance [17]. The expression of HSP90 was significantly increased in the face of pathogenic infection and may perform an important role in the defense of apple. In addition, calcium-dependent MPK4s play a negative regulatory role in SA accumulation and defense responses but a positive regulatory role in growth and development regulation, and their function is conserved among plant species [18]. CDPK is the largest family of calcium-regulated protein kinases in plants. It rapidly senses changes in transient Ca^2+^ signals in plants, recognizes and phosphorylates specific substrates, and then triggers various physiological responses through signal cascade transduction, thereby regulating plant growth and development and response to a variety of stresses. ZmCDPK1 negatively regulates the stress response to cold in maize, and the cotransfection of maize leaf protoplasts with ZmCDPK1 inhibited the expression of the cold-inducible marker gene *Zmerf*3 [19]. CDPK and respiratory burst oxidase homolog (Rboh) are involved in biotic/abiotic stress processes and cell death in plants, and the expression of Rboh plays a key role in the accumulation of ROS, which in turn increases the level of plant defense against pathogens [20]. CDPK expression was significantly downregulated during defense, leading to the hypothesis that CDPK plays a negative regulatory role in the apple defense response to *P. expansum* infection.

The production of ROS is an important way for plants to resist pathogens in their presence. Studies have shown that ROS can be produced in several parts of plant cells, including the cell wall, plasma membrane, peroxisome, and endoplasmic reticulum [21]. In response to external environmental stimuli, a variety of enzymes in the cell wall can produce ROS, and reduced coenzyme II (NADPH) oxidase in the cytoplasmic membrane is the main pathway for hydrogen peroxide (H_2_O_2_) production. The large accumulation of ROS, on the one hand, improve the resistance of plants to pathogens and, on the other hand, can damage the cellular tissue structure of the plant [22]. Thus, in response to the changing external environment, plants have evolved ROS scavenging mechanisms to maintain homeostasis in vivo, mainly through enzymatic and non-enzymatic mechanisms. During *P. expansum* infection, three superoxide dismutases (Cu/Zn-SOD) were identified as differentially downregulated among all differential proteins, together with glutathione peroxidase (GPX). SOD and GPX are enzymes that play key roles in common ROS scavenging mechanisms, and their expression is usually elevated under common biotic, as well as abiotic, stresses. However, SOD and GPX are key enzymes in the common ROS scavenging mechanism. Chang et al. (2009) used the total nonsense transgenic strains AS*-cpGPX* and *Arabidopsis* (At) *GPX7* mutant material located in the chloroplast of *Arabidopsis thaliana* and found that At*GPX1* was responsible for the reduction of H_2_O_2_ under photo-oxidative stress and that Cu/Zn-SOD and Mn-SOD activities were reduced in the AS*-cpGPX* strain under strong light stress, indicating that chloroplast GPX and SOD jointly regulate the balance of O^2−^ and H_2_O_2_ in chloroplasts [23]. We, therefore, hypothesize that SOD and GPX expression is suppressed in the pre-infection period, causing a burst of ROS in response to the infection of *P. expansum*. In the meantime, POD, glutathione transferase (GST), key enzymes in the lignin synthesis pathway, and the accumulation of lignin and callosein the cell wall strengthen the mechanical strength of the cell wall [24]. GST can be induced to be expressed under various biotic or abiotic stresses to enhance plant resistance to adversity and is an important enzyme in the antioxidant defense system as a channel for pathogen dispersal between cells, thus limiting further pathogen invasion into the protoplasm and reproduction in tissues [25]. In the studies of the defense mechanisms between wheat and powdery mildew, the expression of multiple ROS scavenging genes was observed, and GST was shown to contribute to resistance to powdery mildew [26]. The overexpression of GST genes in crops, such as rice and tomato, can improve plant resistance to a variety of biotic stresses [27,28]. These protein changes associated with oxidative stress suggest that the homeostatic levels of intracellular ROS are well maintained during the defense of apples against *P. expansum.*

Induced by pathogenic bacteria, the response is usually accompanied by changes in the content of plant hormones. Ethylene (ET), abscisic acid (ABA), jasmonic acid (JA), salicylic acid (SA), and growth hormones are common hormone species that play a key function in plant defense mechanisms. The JA-, SA-, and ET-mediated signaling pathways are among the metabolic pathways that are associated with disease resistance signals [29,30]. SA is an endogenous phenolic substance that usually accumulates in large quantities in plants after infection by pathogenic fungi, leading to the development of systemic acquired resistance (SAR), which in turn activates the expression of a series of related PR proteins (PR-1, PR-2, and PR-5) [31]. JA and ET usually accumulate in large quantities when pathogens invade, and ET acts as a defense against a wide range of pathogens and acts synergistically with JA in the defense process [32]. In a transcriptomic study of cucumber wilt, inoculation with *Fusarium oxysporum* f. sp. *Cucumerinum* (Foc) the expression of ethylene-related genes was significantly increased in cucumbers inoculated with Foc, and the treatment of cucumbers inoculated with exogenous ET improved resistance to Foc [33]. In the present study, infection by *P. expansum* activated the accumulation of the tryptophan metabolic pathway, the phenylalanine metabolic pathway, the α-linolenic acid metabolic pathway, and other key enzymes for hormone synthesis such as methyltransferase, phenylalanine deaminase, and lipoxygenase in apple cells, inducing hormone synthesis and improving apple resistance.

When a plant is susceptible to a disease, a series of changes in the plant’s defense enzyme system occur that favor plant defense. In this study, the expression of lipoxygenase (LOX), phenylalanine ammonia-lyase (PAL), polyphenol oxidase (PPO), and NADPH-cytochrome P450 reductase (CPR) enzymes were altered to varying degrees (Table 2). PAL is one of the key enzymes in the phenylpropanoid metabolic pathway in plants. Both pathogen infection and pathogenic toxin treatment-induced enhanced PAL activity, and the enhanced enzyme activity was positively correlated with disease resistance. In maize and sugarcane mosaic virus studies (SCMV), the downregulation of ZmPAL expression leads to increased SCMV infection and viral accumulation, as revealed by metabolomic studies, leading to the accumulation of lignin, as well as other metabolites [34]. The exposure of gene-silenced peppers to low temperatures by virus-induced gene silencing revealed a reduction in PAL activity and tolerance to low temperatures, suggesting that PAL may have an important role in the resistance of peppers to low-temperature stress [35]. The LOX-mediated oxylipin synthesis pathway is known as the lipoxygenase pathway, and there are several branching pathways downstream of LOX, such as the allene oxide synthase (AOS) pathway and the hydroperoxide lyase (HPL) pathway. LOX plays an important role when plants are subjected to a variety of biotic and abiotic stresses. ZmLOX10 is an important pest resistance gene in maize that directly resists pests by inducing the production of substances such as JA [36]. Analysis using the *Arabidopsis* 9-LOX deletion mutant LOX1/LOX5 and the oxylipin-insensitive mutant nonresponding to oxylipins (noxy)2-2 and related mutants of brassinosteroids (BRs) showed that oxylipin from 9-LOX induces BR synthesis and signal transduction and activates cell wall responses, such as callus deposition, to limit pathogen invasion and prevent pathogen infection [37]. In a study of walnut resistance to *Xanthomonas arboricola* pv. *juglandis* (*Xaj*), PPO was significantly induced at the site of infection in most walnut varieties, and transgenic strains showed higher PPO activity and lower levels of disease [38]. Niranjan et al. [39] analyzed the role of the isolated PPO gene in downy mildew pearl millet intercropping and showed that the gene accumulated significantly more and faster in downy-mildew-resistant pearl millet seedlings inoculated with *Sclerospora graminicola* PPO mRNA compared to the sensitive control [39]. Wilt and blast are important fungal diseases that are harmful to the health of soybeans and are positively correlated with POD and PPO in the plant, according to experiments [40]. Cytochrome P450 is an important stress-related enzyme in plants. P450s play an important role in plant defense through their involvement in the biosynthesis of plant antitoxins, hormone metabolism, and the biosynthesis of a number of other secondary metabolites [41]. In cytochrome-P450-mediated metabolism of endogenous and exogenous compounds, NADPH–cytochrome P450 reductase (CPR) plays an important role. Most P450 catalytic activity depends on NADPH–cytochrome P450 reductase for electron supply [42]. Thus, in apple, the upregulation of NADPH–CPR expression promoted cytochrome-P450-mediated resistance. Chalcone isomerase (CHI) is a key enzyme in the biosynthetic pathway of flavonoids and flavonoid substances, which have been shown to play an important metabolic role in plant stress resistance. The silencing of the land cotton CHI gene (*GhCHI*) using VIGS technology resulted in the loss of resistance to yellow wilt in cotton, confirming the role of the *GhCHI* gene in cotton resistance to yellow wilt [43]. In a study of the soybean CHI gene, it was found to be expressed in roots, stem, and leaves, and the overexpression of *GmCHI1A* in hairy roots revealed a shortening of spot length and a reduction in free spore germination, indicating that it is regulating the response of soybean to soybean blast [44]. In proteomic studies of apples, the upregulated expression of CHI proteins was likewise identified, which may play a key role in apple defense processes.

A further transmission of disease resistance signals triggers the accumulation of multiple PR proteins in apples, which further resist invasion by pathogenic fungi by increasing cell wall resistance and synthesizing plant antitoxins. Zhang et al. [45] determined that MdPR10-1 and MdPR10-2 promote leaf spot resistance of *Streptomyces* by inhibiting fungal growth, as determined by in vitro experiments [45]. Wang et al. [46] verified the interaction of VmEP1 with MdPR10 through a yeast two-hybrid assay and confirmed the positive regulatory role of MdPR10 in enhancing resistance in apples [46]. The increase in POD activity promoted the accumulation of lignin content and improved fruit disease resistance. PR-2 expression was induced by pathogenic fungi. β-1,3 glucanase can hydrolyze the β-1,3 glycosidic bond, which is an important component of pathogenic fungi, so it is thought that β-1,3 glucan hydrolase can cause pathogen cell lysis and death by hydrolyzing the pathogen cell wall. In vitro functional assays on three grape-derived EGases showed that EGase3 has strong anti-*P. viticola* activity [47]. In *Pectobacterium carotovorum* subsp. *carotovorum* (Pcc) induced higher expression of PR-2 in resistant varieties of tomato than insensitive strains, involved in the defense response of tomato to Pcc [48].

## 5. Conclusions

The cell wall was the first barrier to resist the invasion of *P. expansum*, and the mechanical strength of the apple cell wall was enhanced to resist the invasion of pathogenic fungi through the activation of lignin-synthesis-related pathways. The production of Ca^2+^ channels and the burst of ROS are important signals of the apple immune response, which can improve the defense ability through the production and clearance mechanism of ROS. The activation of PTI and ETI leads to the synthesis of many downstream defense pathways, along with the production of related plant hormones and the expression of defense enzymes. The expression of related PR proteins also directly improves the resistance to pathogens, all of which participated in the response of apples to *P. expansum* infection (Figure 10).

## Figures and Tables

**Figure 1 jof-08-01273-f001:**
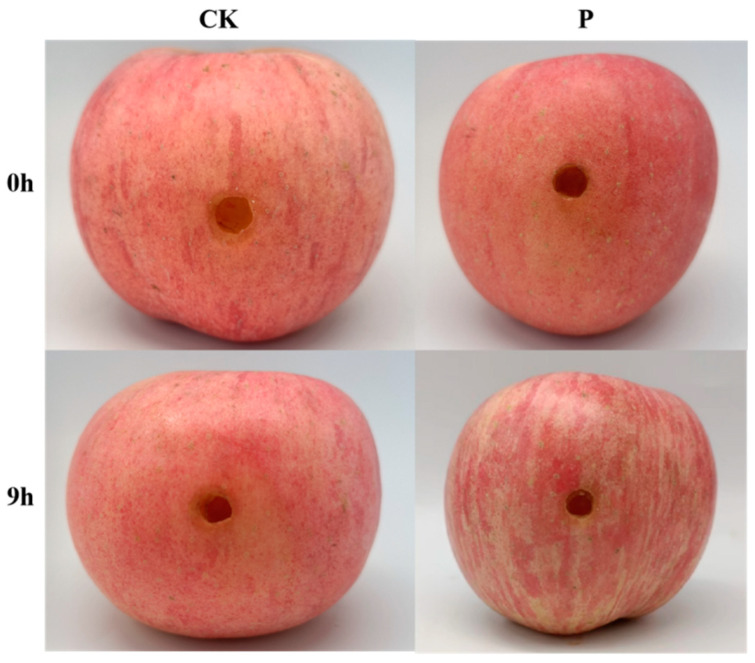
The phenotype of apple inoculated at 0 h and 9 h in control and experimental groups.

**Figure 2 jof-08-01273-f002:**
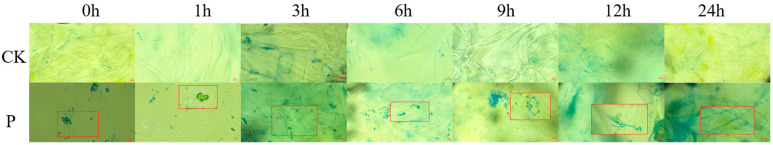
Spore germination status of *P. expansum* at different time points of inoculation.

**Figure 3 jof-08-01273-f003:**
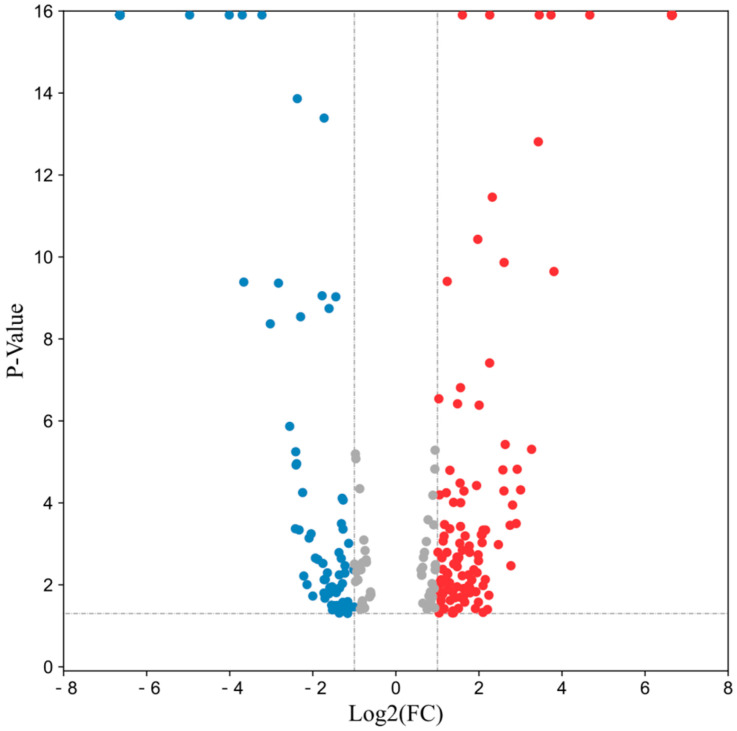
Volcano plot of total differential protein expression. Red represents significantly upregulated differential proteins, and blue represents significantly downregulated differential proteins.

**Figure 4 jof-08-01273-f004:**
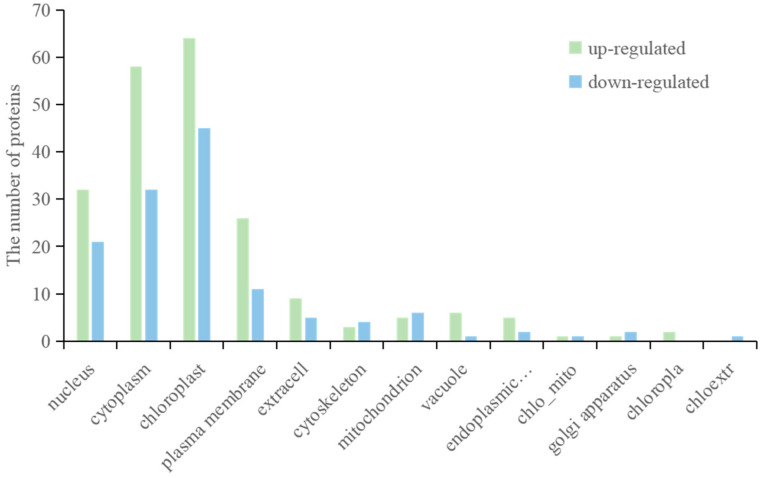
Subcellular localization of DEPs.

**Figure 5 jof-08-01273-f005:**
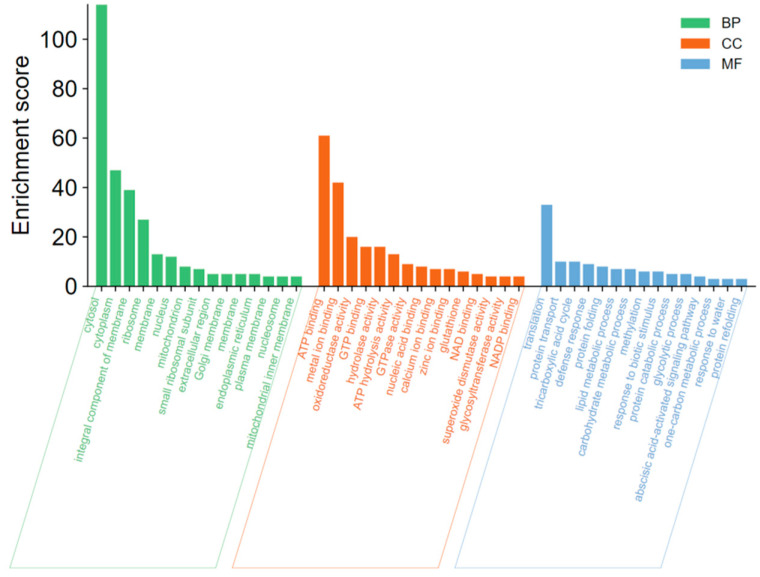
GO enrichment analysis of total DEPs. Divided into three subcategories: biological process, cellular component, and molecular function.

**Figure 6 jof-08-01273-f006:**
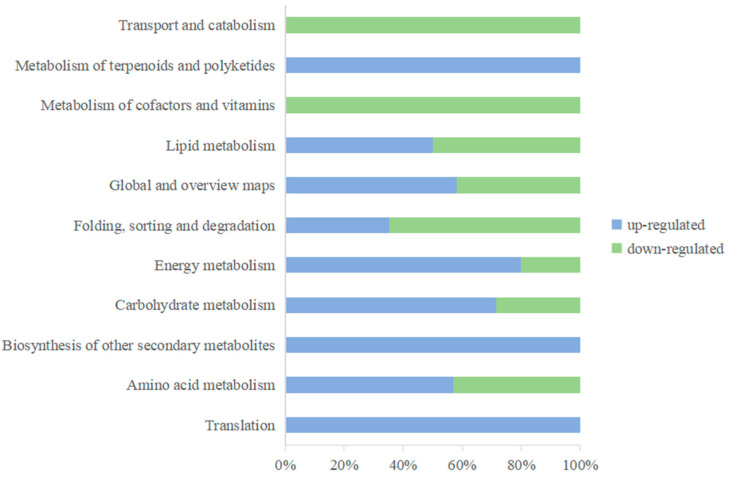
Secondary classification of KEGG functional annotations of DEPs. Blue represents upregulated proteins, and green represents downregulated proteins.

**Figure 7 jof-08-01273-f007:**
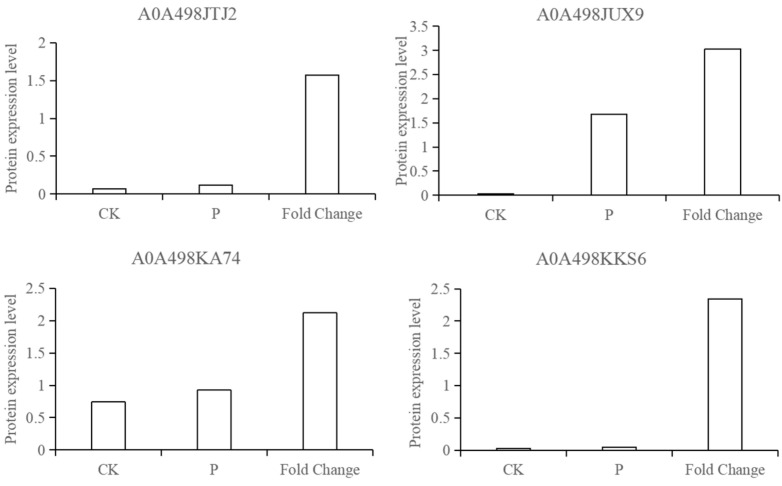
PRM verification of candidate proteins. CK and P represent the protein expression in PRM verification, and fold change is from the proteomic data.

**Figure 8 jof-08-01273-f008:**
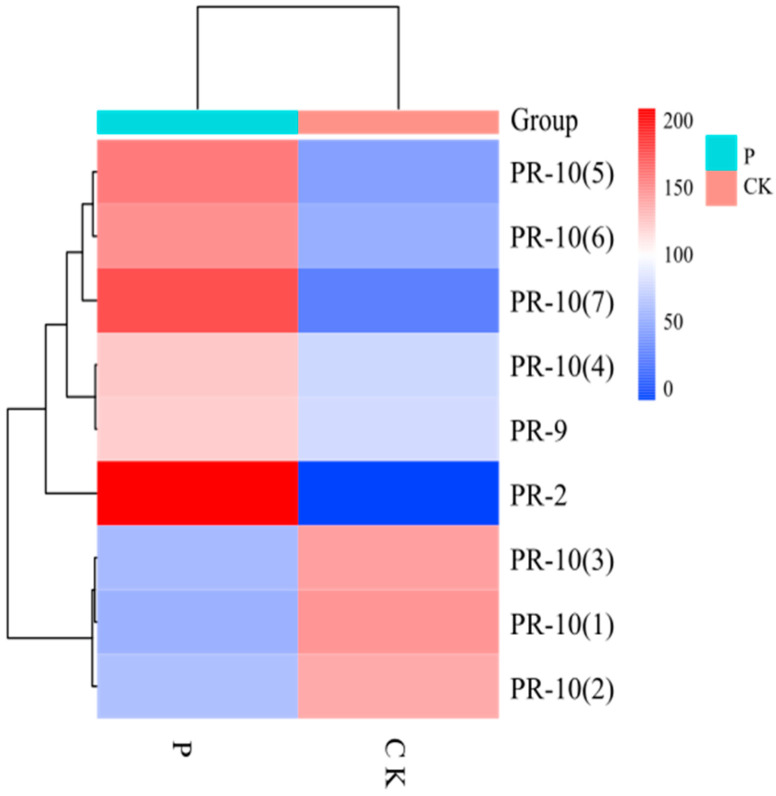
Clustering of pathogenesis-related proteins expression. The depth of color indicates how much different proteins are expressed in the sample.

**Figure 9 jof-08-01273-f009:**
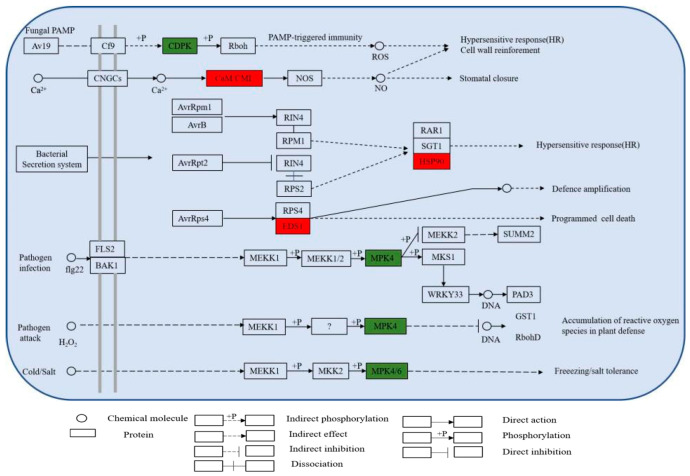
Part of the metabolic pathways involved in defense responses in apples. Red indicates upregulated DEPs, and green indicates downregulated DEPs.

**Figure 10 jof-08-01273-f010:**
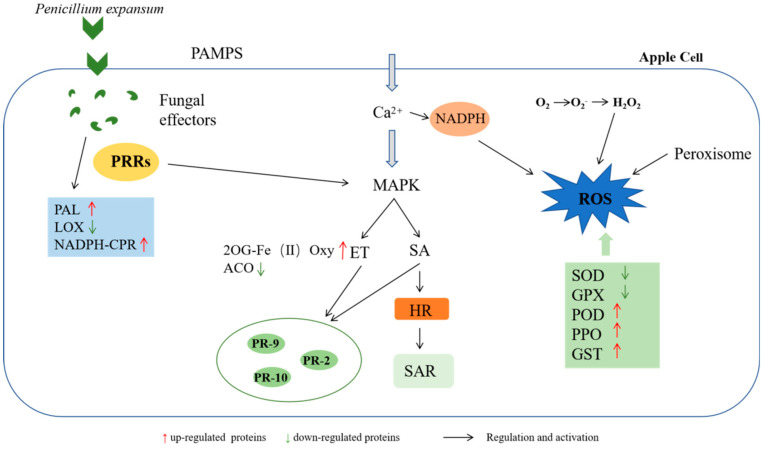
Map of defense mechanisms in response to *P. expansum* infection in apples.

**Table 1 jof-08-01273-t001:** Important metabolic pathways related to apple defense and the number of DEPs.

Pathways	The Number of Upregulated DEPs	The Number of Downregulated DEPs
Plant-pathogen interaction	3	2
Pyruvate metabolism	4	2
Oxidative phosphorylation	4	0
Alpha-linolenic acid metabolism	1	1
Phenylpropanoid biosynthesis	5	0
Flavonoid biosynthesis	1	0
Glutathione metabolism	1	2
MAPK signaling pathway—plant	0	1
Ubiquitin-mediated proteolysis	0	1

**Table 2 jof-08-01273-t002:** Expression levels of key enzymes in apple response to *P. expansum* infection.

Proteins	Description	FC	*p*-Value
A0A498K3F2	Superoxide dismutase [Cu-Zn]	0.609	0.00244296
A0A498K8S2	Superoxide dismutase [Cu-Zn]	0.612	0.00281705
A0A498INN8	Superoxide dismutase	0.645	0.01964518
A0A498JNW6	Glutathione peroxidase	0.292	8.84 × 10^−10^
S4UL39	Lipoxygenas	2.941	1.54 × 10^−7^
A0A498HL71	Methyltransferase	2.158	0.00218763
A0A498JZC5	Phenylalanine ammonia-lyase	2.949	9.91 × 10^−5^
A0A498K3I5	Chalcone isomerase	100	1.25 × 10^−16^
A0A498HJB2	NADPH–cytochrome P450 reductase	100	1.25 × 10^−16^
A0A498IC26	NADPH–cytochrome P450 reductase	2.202	0.00085584
Q93XM8	Polyphenol oxidase 2	100	1.25 × 10^−16^
A0A540LHP9	Peroxidase	1.855	6.51 × 10^−5^
A0A498HW70	Glutathione transferase	1.664	0.00087862

## Data Availability

The datesets generated during and/or analysed during the current study are available from the corresponding author on reasonable request.

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
