# Peer review of "Proteomic Analysis of Apple Response to *Penicillium expansum* Infection Based on Label-Free and Parallel Reaction Monitoring Techniques"

_jof, 2022, doi:10.3390/jof8121273_

Round 1
Reviewer 1 Report
The manuscript entitled “Proteomic analysis of apple response to Penicillium expansum infection based on label-free and PRM techniques.” investigated the apple response to Penicillium expansum infection by label-free and PRM techniques. The differential expression of enzymes related to apple response to pathogen invasion was found in this research at the key time point (9 h). It reveals the mechanisms of apple response at the proteomic level at the 9 h of P. expansum infection. This study is meaningful and well written. I recommend that it should be accepted for publication after minor revision. I propose that the content below should be revised carefully:
1. It is better to write the full name of the word in the title without abbreviations, PRM should read as “parallel reaction monitoring” Please revise it.
2. Line 20 - 21: “This study reveals the mechanism of apple response at the proteomic level during the 9 h of P. expansum infection.” should read as “This study reveals the mechanisms of apple response at the proteomic level at 9 h of P. expansum infection.”
3. Line 51-56: “In the adaptation of fungi to environmental pH, pacC plays an important function. chen et al.(2017) found through proteomic studies that PePacC can act as an effector for a variety of target proteins and play an important role in extending the virulent synthesis of P. expansum. Prusky et al. (2004) found from studies on apples and citrus that PePG1 expression was strongly correlated with environmental pH and that environmental acidification was an important cause of enhanced pathogenicity of pathogenic fungi [vi]” Should read as “pacC plays an important role in the adaptation of fungi to environmental pH. Chen et al.(2017) found that PePacC can act as an effector for a variety of target proteins and play an important role in extending the virulent synthesis of P. expansum; Prusky et al. (2004) found that that PePG1 expression was strongly correlated with environmental pH on apples and citrus and that environmental acidification was important in enhancing pathogenicity of pathogenic fungi.” The name of the gene should be italicized. Check the manuscript carefully.
4. Line 154: “At different time points of infection, samples were taken and sections were made.” “Sections” in this sentence is confused and inappropriate here in my opinion, I think it should be “wounds”, revise it please. This sentence should read as “Samples were taken and wounds were made at different time points during P. expansum infection.” Please pay attention to the grammars.
5. Line 424 and 431: The phrase “in vitro” should be italicized. Check the manuscript carefully.
6. I think there is a mistake in Fig 10, PRRS should be “PRRs”, this means “Pattern recognition receptors”. It’s better to add more details in Figure 9 and Figure 10, such as the meaning of some straight lines, dotted lines and some special symbols in Figure 9, the same thing in Figure 10.
Author Response
Dear reviewer,
Thank you for your decision and constructive comments on my manuscript. We have carefully considered the suggestion of reviewer and make some changes. We have tried our best to improve and made some changes in the manuscript.
The revision notes (point-to-point) are as follows:
Reviewer #1:The manuscript entitled “Proteomic analysis of apple response to Penicillium expansum infection based on label-free and PRM techniques.” investigated the apple response to Penicillium expansum infection by label-free and PRM techniques. The differential expression of enzymes related to apple response to pathogen invasion was found in this research at the key time point (9 h). It reveals the mechanisms of apple response at the proteomic level at the 9 h of P. expansum infection. This study is meaningful and well written. I recommend that it should be accepted for publication after minor revision. I propose that the content below should be revised carefully:
- It is better to write the full name of the word in the title without abbreviations, PRM should read as “parallel reaction monitoring” Please revise it.
Response: Please, the title has been amended according to the suggestion in the revised manuscript.
- Line 20 - 21: “This study reveals the mechanism of apple response at the proteomic level during the 9 h of P. expansum infection.” should read as “This study reveals the mechanisms of apple response at the proteomic level at 9 h of P. expansum infection.”
Response: Please, this point has been revised as recommended in line 22, 23 of the revised manuscript.
- Line 51-56: “In the adaptation of fungi to environmental pH, pacC plays an important function. chen et al. (2017) found through proteomic studies that PePacC can act as an effector for a variety of target proteins and play an important role in extending the virulent synthesis of expansum. Prusky et al. (2004) found from studies on apples and citrus that PePG1 expression was strongly correlated with environmental pH and that environmental acidification was an important cause of enhanced pathogenicity of pathogenic fungi [vi]” Should read as “pacC plays an important role in the adaptation of fungi to environmental pH. Chen et al.(2017) found that PePacC can act as an effector for a variety of target proteins and play an important role in extending the virulent synthesis of P. expansum; Prusky et al. (2004) found that that PePG1 expression was strongly correlated with environmental pH on apples and citrus and that environmental acidification was important in enhancing pathogenicity of pathogenic fungi.” The name of the gene should be italicized. Check the manuscript carefully.
Response: Please, this point has been revised as recommended in line 54-59 of the revised manuscript.
- Line 154: “At different time points of infection, samples were taken and sections were made.” “Sections” in this sentence is confused and inappropriate here in my opinion, I think it should be “wounds”, revise it please. This sentence should read as “Samples were taken and wounds were made at different time points during expansum infection.” Please pay attention to the grammars.
Response: Please, this point has been revised as recommended in line 157 and 158 of the revised manuscript.
- Line 424 and 431: The phrase “in vitro” should be italicized. Check the manuscript carefully.
Response: Please, this point has been revised as recommended in line 435 and 442 of the revised manuscript.
- I think there is a mistake in Fig 10, PRRS should be “PRRs”, this means “Pattern recognition receptors”. It’s better to add more details in Figure 9 and Figure 10, such as the meaning of some straight lines, dotted lines and some special symbols in Figure 9, the same thing in Figure 10.
Response: Please, “PRRS” has been revised as “PRRs”. We have added the notes in Figure 9 and Figure 10 as recommended in the revised manuscript.

Reviewer 2 Report
The manuscript entailed "Proteomic analysis of apple response to Penicillium expansum infection based on label-free and PRM techniques" by Xu et al described their observations on the mechanism of apple response at the proteomic level during the 9 h of P. expansum infection. However, I have a few concerns listed below.
1. The resolution in figure 5 and figure 9 are really too low to see.
2.The author observed the infection of the apple tissues in 6h, 9h and 12h, first the microscope images scale bar is needed; second, are there any differences in the degree of penetration about P. expansum in apple tissues in different time point?
3. In line 103, the samples were taken at the wound site with a sterile scalpel. The wounds will also cause the defense response, did the author tested whether the wounds will cause additional effect on the infection of P. expansum?
4. In figure 4, are the up-regulated proteins have the similar subcellular localization with the dowm- regulated proteins?
5. Figure 7 needs Y axis.
6. EDS1 is involved in the SA pathway, is the invasion of P. expansum caused the activation of SA pathway? Other SA pathway proteins identified?
7. The identified disease resistance proteins need further experiments to confirm. For example, knockout or over-expression, to confirm their function.
Author Response
Dear reviewer,
Thank you for your decision and constructive comments on my manuscript. We have carefully considered the suggestion of reviewer and make some changes. We have tried our best to improve and made some changes in the manuscript.
The revision notes (point-to-point) are as follows:
Reviewer #2:The manuscript entailed "Proteomic analysis of apple response to Penicillium expansum infection based on label-free and PRM techniques" by Xu et al described their observations on the mechanism of apple response at the proteomic level during the 9 h of P. expansum infection. However, I have a few concerns listed below.
- The resolution in figure 5 and figure 9 are really too low to see.
Response: Please, we have changed figure 5 and figure 9 with more clear figures in the revised manuscript.
- The author observed the infection of the apple tissues in 6h, 9h and 12h, first the microscope images scale bar is needed; second, are there any differences in the degree of penetration about expansum in apple tissues in different time point?
Response: Thank you for your suggestion. In the photos of spore germination, the microscope scale bars are added now; The infection degree of P. expansum in apple tissues at different time points are explained in detail in lines 158-167 of the revised manuscript.
- In line 103, the samples were taken at the wound site with a sterile scalpel. The wounds will also cause the defense response, did the author tested whether the wounds will cause additional effect on the infection of P. expansum?
Response: Please, the control group was injected with normal saline at the wounds, and the same amount of P. expansum spores suspension was used as the experimental group The two groups were treated in the same way (i.e wounds were made when we applied the saline and the P. expansum spore suspension). Therefore, the major difference is the treatment as both group were wounded. Therefore, wounds were existed in CK and treated groups, the defense effect between the two groups were compared irrespective of the effect of the wounds created.
- In figure 4, are the up-regulated proteins have the similar subcellular localization with the down- regulated proteins?
Response: Please, we analyzed the subcellular localization of up-regulated and down-regulated proteins, and found that they were located in similar positions. We also revised this point as recommended in figure 4 of the revised manuscript.
- Figure 7 needs Y axis.
Response: Please, this point has been revised as recommended in figure 7 of the revised manuscript.
- EDS1 is involved in the SA pathway, is the invasion of expansum caused the activation of SA pathway? Other SA pathway proteins identified?
Response: Please, PTI can indirectly activate EDS1 and PAD4, further promote the expression of PAL and ICS, leading to the gradual accumulation of SA, in this study, PAL is up-regulated and involved in the activation of SA signaling pathway.
- The identified disease resistance proteins need further experiments to confirm. For example, knockout or over-expression, to confirm their function.
Response: Please, protomic sequencing compares peptide sequences with known databases to obtain differentially expressed proteins. This study screened some resistant proteins through bioinformatics analysis and a large number of literature support. We will use methods such as over- expression and knockout to verify the function of these proteins in the subsequent experimental plan.
